# A qualitative study examining the health system's response to COVID-19 in Sierra Leone

Hana Stone[1‡], Emma Bailey[1,2‡], Haja Wurie[3], Andrew J. M. Leather[1], Justine I. Davies[4,5,6], Håkon A. Bolkan[7,8,9], Stephen Sevalie[3,10,11], Daniel Youkee[2,10], Divya Parmar[1]*

1 King's Centre for Global Health and Health Partnerships, Department of Population Health Sciences, School of Life Course and Population Sciences, King's College London, London, United Kingdom, 2 King's Sierra Leone Partnership, Connaught Hospital, Freetown, Sierra Leone, 3 College of Medicine and Allied Health Sciences, University of Sierra Leone, Freetown, Sierra Leone, 4 Institute of Applied Health Research, University of Birmingham, Birmingham, United Kingdom, 5 Centre for Global Surgery, Department of Global Health, Stellenbosch University, Cape Town, South Africa, 6 Medical Research Council/Wits University Rural Public Health and Health Transitions Research Unit, Faculty of Health Sciences, School of Public Health, University of the Witwatersrand, Johannesburg, South Africa, 7 CapaCare, Freetown, Sierra Leone, 8 Institute of Clinical and Molecular Medicine, Norwegian University of Science and Technology, Trondheim, Norway, 9 Department of Surgery, St. Olavs Hospital HF, Trondheim University Hospital, Trondheim, Norway, 10 Case Management Pillar, National COVID-19 Emergency Response Centre, Freetown, Sierra Leone, 11 34th Military Hospital, Wilberforce, Freetown, Sierra Leone

‡ HS and EB are joint first authors on this work.
* divya.parmar@kcl.ac.uk

**Data Availability Statement:** Data are available at OSF: https://osf.io/qc3z8/ doi: DOI 10.17605/OSF.IO/QC3Z8.

## Abstract

The paper examines the health system's response to COVID-19 in Sierra Leone. It aims to explore how the pandemic affected service delivery, health workers, patient access to services, leadership, and governance. It also examines to what extent the legacy of the 2013–16 Ebola outbreak influenced the COVID-19 response and public perception. Using the WHO Health System Building Blocks Framework, we conducted a qualitative study in Sierra Leone where semi-structured interviews were conducted with health workers, policymakers, and patients between Oct-Dec 2020. We applied thematic analysis using both deductive and inductive approaches. Twelve themes emerged from the analysis: nine on the WHO building blocks, two on patients' experiences, and one on Ebola. We found that routine services were impacted by enhanced infection prevention control measures. Health workers faced additional responsibilities and training needs. Communication and decision-making within facilities were reported to be coordinated and effective, although updates cascading from the national level to facilities were lacking. In contrast with previous health emergencies which were heavily influenced by international organisations, we found that the COVID-19 response was led by the national leadership. Experiences of Ebola resulted in less fear of COVID-19 and a greater understanding of public health measures. However, these measures also negatively affected patients' livelihoods and their willingness to visit facilities. We conclude, it is important to address existing challenges in the health system such as resources that affect the capacity of health systems to respond to emergencies. Prioritising the well-being of health workers and the continued provision of essential routine health

**Funding:** This research was funded by the National Institute of Health Research (NIHR) Global Health Research Unit on Health System Strengthening in Sub-Saharan Africa (GHRU 16/136/54) using UK aid from the UK Government to support global health research. DP is funded by the GACD-MRC IMPACT Grant and DY by NIHR (GHR:17:63:66) using UK aid from the UK Government to support global health research. The views expressed in this publication are those of the author(s) and not necessarily those of the funders or the UK government. The funders had no role in study design, data collection and analysis, decision to publish or preparation of the manuscript.

**Competing interests:** The authors have declared that no competing interests exist.

services is important. The socio-economic impact of public health measures on the population needs to be considered before measures are implemented.

## Introduction

At the time of this study, there were 2560 confirmed COVID-19 cases reported and 76 deaths [1]. Reported infection and mortality rates attributable to COVID-19 in Sierra Leone have been a fraction of the predicted modelled scenarios [2] published at the start of the pandemic. This could be attributed to several factors including but not limited to the COVID-19 response in Sierra Leone being implemented through lessons learnt lens.

Sierra Leone is no stranger to disease outbreaks; COVID-19 is the second major epidemic to affect Sierra Leone in the last ten years. In 2013–16 West Africa experienced a major Ebola outbreak. After an initially delayed reaction [3] the Ebola response in West Africa embodied a complex and collective effort costing more than 3.6 billion USD [4] and involving the collaboration of national staff with 58 international teams from 40 organisations [5]. Key criticisms of the international approach included unaccountable leadership, the failure of rapid response mechanisms for case detection and the absence of global alerts for knowledge sharing [6]. In terms of mortality Sierra Leone registered 3956 Ebola deaths [7] and distortion of resources and focus on Ebola [8] was linked to an 11% reduction in facility births, 34% increase in hospital maternal mortality, 24% increase in stillbirth rate [9] and 2819 excess deaths attributable to HIV/AIDS, Malaria & TB [10].

In contrast to Ebola, Sierra Leone's COVID-19 response was nationally led and implemented rapidly. The first case of COVID-19 was reported on the 30[th] of March 2020 with lockdown measures, inter-district travel bans, and curfews, all introduced shortly after in the first and second weeks of April 2020 [11]. The national response was also designed to ensure the continuity of essential care services. In addition, awareness of COVID-19 was high, with strong associations between increased knowledge and practices [12]. Due to this, national non-COVID-19 health service utilisation was initially affected with a decrease in facility admissions to medical and surgical beds although these decreases were less severe than those seen during the Ebola epidemic and less than decreases seen globally [13]. To a large extent the perceptions of communities during the Ebola outbreak towards health services revolved around conspiracy theories, fear, and mistrust [8, 14] stemming from suspicion about the involvement of international teams, the origin of the virus and mistrust in healthcare delivered in treatment centres outside of familiar facilities. Thus, health workers experienced isolation, stigmatisation, sadness, and a loss of coping mechanisms [15].

Studies based in Sierra Leone during the COVID-19 pandemic have reported a high public risk perception, a misunderstanding of mortality rates [12], and the negative impact of national restrictions on household finance and social freedoms [8]. However, there remains a limited understanding of the health system's response to the COVID-19 pandemic in Sierra Leone. This qualitative study contributes to the evidence base by examining the early response to COVID-19, covering the first 9 months of the pandemic. We used the World Health Organisation's (WHO) Health Systems Building Blocks framework [16] to examine how the pandemic affected service delivery, health workforce, information sharing, access to medicines and supplies, financing, and leadership and governance. In addition, we study the patient experience of accessing and receiving services and explore how the previous Ebola epidemic may have shaped the current response. The Ebola epidemic caused substantial reductions in major surgeries and routine maternity care, while emergency care such as caesarean deliveries

were more resilience [17, 18]. We, therefore, focus on the provision of emergency and routine care, specifically, maternal, and surgical services.

## Methods

This qualitative study is part of a larger body of work examining the health system response to COVID-19 in Sierra Leone; it includes analysis of hospital admissions data, as part of a mixed-methods study published elsewhere [13].

### Study design

Data was collected through semi-structured interviews with health workers, patients, and policymakers during October—December 2020. A total of five health facilities in two of the 16 districts of Sierra Leone were purposely selected (Table 1). Study sites were selected in consultation with local researchers to be representative of those providing health services for the population and by considering travel restrictions. Primary Health Units (PHUs) were selected with the highest case volume as per the latest Service Availability and Readiness Assessment 2017 [19].

We interviewed health staff, patients, and key informants (KIs). Health staff were identified by the facility heads depending on their availability and experience of working in the surgical/maternity wards or those involved with the management of the facility. In each facility, at least two health staff and one adult patient were interviewed. Additional senior staff were interviewed from secondary and tertiary hospitals who has a more district-wise perspective. The patients were approached by the staff in maternity and surgical wards and if they agreed to be interviewed, they were later interviewed at their home or a convenient location. KIs were identified by the local research team. Two were involved in the national COVID-19 response holding national planning and policy roles and one was involved in providing technical support from an international organisation (Table 2).

Interviews were conducted by two experienced and trained Research Assistants (RAs). The RAs received a -day training consisting of mock interviews, transcription training and a pilot interview. The interviews were conducted in English or Krio, depending on participant preference. Interviewers ensured understanding of the questions by repeating or rephrasing as needed. Interviews were audio-recorded with permission and transcribed by the RAs. Krio interviews were translated during transcription. A sample of transcriptions (30%) were crossed checked. The last author cross-checked the English transcriptions and the RAs checked each other's Krio transcriptions. Twenty-three interviews were conducted in person; one KI interview was conducted by phone. All respondents provided written signed informed consent.

The interview guide was based on the WHO's Health System Building Blocks [20] and asked questions on changes to service provision, health staff, finances of health facilities, access to information and medical supplies, and leadership. Patients were additionally asked about experiences of accessing healthcare and KI responses provided organisational and national

**Table 1. Facility characteristics.**

| District | Facility type | Area type |
|---|---|---|
| One | Secondary hospital | Semi-urban |
| | Primary health unit | Rural |
| Two | Tertiary hospital | Urban |
| | Secondary hospital | Urban |
| | Primary health unit | Urban |

**Table 2. Interviewee demographics (n = 24).**

| Role | Male | Female | Total |
|---|---|---|---|
| *Health workers* (n = 16) | | | |
| Doctor | 2 | 1 | 3 |
| Surgeon | 3 | 0 | 3 |
| Midwife | 0 | 1 | 1 |
| Maternal & Child Health Aide | 0 | 1 | 1 |
| Nurse | 0 | 1 | 1 |
| Matron | 0 | 4 | 4 |
| Finance Officer | 0 | 2 | 2 |
| Deputy in charge | 1 | 0 | 1 |
| Patients (n = 5) | 2 | 3 | 5 |
| Key informants (n = 3) | 3 | 0 | 3 |

perspectives on COVID-19 health systems response. All respondents were asked to reflect on their experience and learnings from the Ebola epidemic. Interview guides were piloted in a health facility by the RAs as part of their training. Interview guides are included as S1 File.

Ethical approval was granted by the Sierra Leone Ethics and Scientific Review Committee and the Regional Committee for Medical and Health Research Ethics in Norway (2020/155388). Letters of support were also received from the Ministry of Health and Sanitation and the Ministry of Defence. Additional information regarding the ethical, cultural, and scientific considerations specific to inclusivity in global research is included in the S1 Checklist.

## Data analysis

Transcripts were read multiple times for immersion before thematic analysis was conducted. We used both inductive and deductive approaches [21, 22] to identify themes and sub-themes. The deductive approach identified themes based on the WHO Health System framework [20] which matched the design of the interview questions. Additional themes about the patient experience and learning from Ebola, not captured in the WHO Health System framework, were identified inductively. Coding was conducted by HS using NVivo 12. Initial codes were reviewed and agreed by three authors (HS, EB and DP) and the final themes and subthemes were decided in consultation with other authors. Theme saturation was reached during data analysis when all transcripts had been read multiple times and coding completed. We believe if it were feasible to interview more patients and national response KIs, additional new themes outside of the thematic framework may have emerged providing alternative insights to those presented.

## Results

We conducted interviews with health workers, key informants (KIs) and patients and identified 12 overarching themes: nine related to the six WHO Health System Building Blocks [20], two related to patient experiences of accessing healthcare, and the last on Ebola (Fig 1).

### Service delivery

**1. Changes to services.** There were mixed opinions about the changes to routine services. Elective surgery was reported to be the most affected along with accident and emergency departments and specialist outpatient services although this changed over time as patient

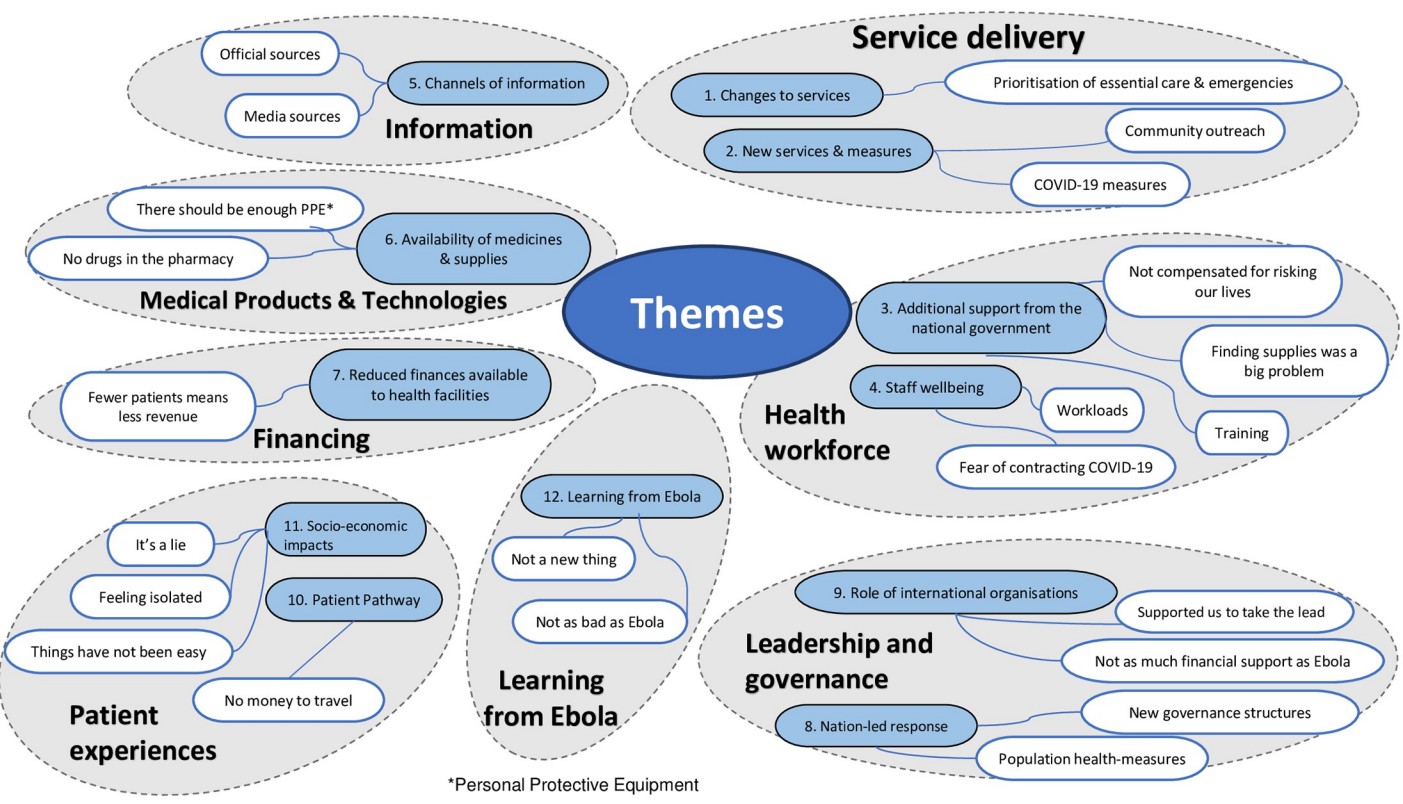

**Fig 1. Themes and sub-themes.**

attendances fluctuated. Reduced patient numbers were mainly associated with the early days of the outbreak. In contrast, other services were reportedly functioning well—due to efficient and organised systems such as screening patients for COVID-19 symptoms before entering a facility.

*Prioritisation of essential care and emergencies.* In response to a national policy decision, elective surgeries were halted across facilities to prioritise the continuation of essential and emergency surgery [13]. In part, this caused confusion as staff and patients were not aware of the types of services available as the policy was interpreted differently between facilities.

> *"At the height of the covid, everyone was afraid. So only emergency patients were operated on in the surgical department. All elective cases were put on hold."*

> *(Senior Doctor)*

> *"When they go to other hospital like Lumley hospital, Rokupa hospital and all other hospital they send them back here . . . they send them to Connaught."*

> *(Senior Surgeon)*

**2. New services and measures.** New services and measures were adopted during COVID-19 including those taken by formal health workers for example the treatment of positive COVID-19 patients within facilities alongside routine care, as well as external community engagement by community health workers.

*Community outreach.* When health facilities felt that patients were not accessing in-hospital services because of fear of contracting COVID-19 a small number of health providers started community outreach programmes–for HIV/TB services this was a deliberate policy decision whereas for other services it was unclear if this was implemented due to individual initiative or directed by the Ministry of Health and Sanitation (MoHS). Community health workers were given information by facility staff to disseminate and used these community visits to build trust and reassure people that facilities were still providing healthcare and that facilities were not a site of infection.

*"We gave health talks through the CHWs [community health workers]. Some people says that the CHWs don't work, but I swear to God, mines do work. . .Some [patients] will come here and tell us that it was that CHW who told so and so. So, that is why they [patients] were at the facility."*

*(Senior Nurse in Charge)*

*COVID-19 measures.* Health workers reported following protocols to identify positive COVID-19 patients through triage and testing. Patients were screened on entry and those with COVID-19 symptoms were isolated before being tested and referred to treatment centres if necessary.

*"If we get a suspect, we have an isolation unit, infection disease unit. We take them there direct. They don't even enter here [facility]. From where [triage] they do the screening- out-side- take them directly to the IDU [Infectious Disease Unit]."*

*(Senior Matron)*

Additional Infection Prevention and Control (IPC) measures were also introduced. Health workers, patients, and visitors were not permitted to enter facilities without masks and regular hand washing or using hand sanitiser was encouraged. Staff were required to wear Personal Protective Equipment (PPE) when examining patients.

*"The outbreak was not like what we were seeing before. So now we examine patient when we are fully dressed in our PPE's our gowns our facemask gloves and everything, we examine patient now."*

*(Senior Doctor)*

## Health workforce

**3. Additional support from the national government.**   Health workers highlighted compensation and additional work required to source supplies as aspects where the national government could have better supported them during the pandemic. They reported increased risks associated with treating COVID-19 patients and purchasing their own equipment in the absence of requisitions. Health workers reported that training was widely accessible although it created an additional burden.

*Not compensated for risking our lives.* Pay, bonus and service benefit issues affected staff morale and service delivery. Health workers expected that they should have been paid extra for working with suspected and positive COVID-19 patients. In 2020, health workers took strike

action in response to delays and confusion about salary payments and additional financial "*risk bonuses*".

*"Definitely after risking your life with the covid patients, and you are not paid for your work, the morale force will be down. There is a pull down of morale. Especially for staff working at the Covid unit".*

*(Senior Doctor)*

*Finding supplies was a big problem.* Lack of supplies increased pressure and workload on health workers, who needed to do more work to source equipment through formal and informal channels.

*"This is the big problem that we face. Even recently, the requisition I told you about. . .it almost took a week. . . I had to write again, make a requisition. If you want, I can even show you the paper, but they did not supply. I included cotton wool, blood syrup and all that is lacking."*

*(Senior Deputy)*

*Training.* Many health workers had received IPC training, delivered during the Ebola period, and received refresher training at the start of the pandemic. All facilities where we conducted interviews had received training, directly or through a cascade model, in managing risk around COVID-19 infection, transmission, and treating patients safely.

*"I went for the training, come back, and cascaded it. That of my other colleague also went for training, she came and also cascaded it."*

*(Senior Nurse in Charge)*

**4. Staff wellbeing.** The pandemic had an impact on the general well-being of the health workers.

*Fear of contracting COVID-19.* Health workers considered themselves as a high-risk population and therefore were more vulnerable to COVID-19 infection, similar to the Ebola outbreak. Owing to the initial uncertainty around COVID-19, many felt stressed and were afraid of contracting COVID-19. Reports of COVID-19 outbreaks in health facilities where departments or facilities were closed further increased their anxiety.

*"Nearly everyone was stress you know because one you think about the people that have infected, then the whole general hospital because of the highly nature contagious in the diseases."*

*(Senior Surgeon)*

*Workloads.* All health workers reported an overall decrease in patients accessing services in the initial phase of COVID-19, leading to a reduced workload for most.

*"[caseloads] reduced. Like the specialist outpatient, were not seeing cases. Even the general entrance, accidents and emergencies reduced."*

*(Senior Matron)*

However, because of fewer health workers, workloads were also reported to increase for some teams. Staff rotation was changed so that fewer health workers were present at any given time, and some were also deployed elsewhere to run COVID-19-related services and IPC training.

*"Yah, that affected us negatively somehow because if you are in the unit where we normally have like three to four doctors working earlier, we have reduce the two because the other two will need to go for, e.g. for IPC training and the rest. Of course, the workload will be too much on those on ground."*

*(Senior Doctor)*

## Information

**5. Channels of information.** Health workers relied on official and unofficial sources of information on COVID-19. The official sources existed within the public health systems while media included radio/TV/print and messaging platforms. There were mixed perceptions of the amount of communication from the national government to health facilities.

*Official sources.* Within most facilities, COVID-19 response teams were set up, involving senior management and key department leads who met frequently to review risks—cascading decisions and information through to facility departments. Health workers received information about COVID-19 in meetings and updates from their facilities or through official hospital WhatsApp groups.

*"Senior matron, they meet every. . .once a week for updates on whatever goes on in the hospital. Then they have the covid response team at Connaught, that also meet, they meet on every Monday and give report. They meet and discuss issues."*

*(Senior Matron)*

*"Yes, they have group on WhatsApp, the SLMDA [Sierra Leone Medical and Dental Association], the JUDASAIL [Junior Doctors Association in Sierra Leone] and there is one other group."*

*(Senior Doctor)*

*Media sources.* Sources of information external to the hospital facilities were also important. Health workers actively sought information from WhatsApp, the internet, radio, and TV.

*"I joined [a] series of groups, so whenever they receive any information on the COVID-19, they will send it to our group. I also get information from reading the news from WhatsApp. I also get it from listening to news on radios, and even over television. So nationally I do get information."*

*(Senior Deputy in Charge)*

## Medical products & technologies

**6. Availability of medicines and supplies.** Shortages of equipment were compounded by COVID-19 resulting in a lack of PPE and pharmacy stock-outs in some facilities, although not all.

*There should be enough PPE.* Availability of PPE was reported as mixed across and within facilities during the start of the pandemic. There seemed to be some rationing of PPE and it was perceived that urban and larger facilities had better availability of PPE than rural areas and smaller facilities. Even within facilities, surgery departments were perceived to have fewer shortages of PPE than maternity wards. In facilities that faced shortages, health workers resorted to buying their own and sometimes they also asked patients to buy items.

*"Because when they say PPE, it should be available enough. . .when you want to use it, it [should be] readily available."*

*(Senior Matron)*

*"What we did at times we ask the patients to go and buy boxes of gloves."*

*(Senior Midwife)*

*"Later on, the government then intervened and sent those IPCs- gloves, masks, the N95s [surgical masks]."*

*(Senior Doctor)*

*No drugs in the pharmacy.* Most staff reported that shortages of medicines became worse during the pandemic; patients often had to buy these from private pharmacies or outside traders. Both health workers and patients reported worsening pharmacy stock-outs suggesting intra-district travel bans and a reduction in trade as a potential cause.

*"When there are no drugs in the pharmacy to give to the patient, then the burden will lie on the patient."*

*(Senior Doctor)*

*"Like for me I needed some drugs I went there for this operation and certain drugs that they prescribed for me it wasn't available."*

*(Patient)*

### Financing

**7. Reduced finances available to health facilities.** Staff reported that the financial structure of their facilities did not change during COVID-19, the main source of revenue remained out-of-pocket payments.

*Fewer patients mean less revenue.* In the absence of universal healthcare, most facilities generate revenue from patients paying for healthcare except for patients eligible for the Free Healthcare Initiative (FHCI). The FHCI is a government scheme providing free care to pregnant women, lactating mothers, and children under five [23], which later expanded to cover Ebola survivors and some people with disabilities. During the pandemic, because of lower patient loads, many facilities experienced a reduction in revenue. Facilities did not receive any additional funds from the MoHS to cover this gap.

*"We are not getting enough revenue. This is because of the low patient turnout. We spend more than what we receive. . .Most often people are under oxygen, so there should be twenty-*

*four hours light. And when there is no light, we have to put on the generator. The generator uses lots of fuel because it is very big. So, the hospital was running under loss."*

*(Finance Officer)*

*"If the electricity is off, we pulled from the revenue account and buy, if thermometer get spoiled, we buy, everything they used all the instruments blood bags we take care of every other thing, as long as the money is within the range, we will solve it quickly before disgrace will come, that what we use the revenue money for."*

*(Senior Finance Officer)*

## Leadership and governance

**8. Nation-led response.** The response to COVID-19 encompassed population health measures, governance structures, and international organisations. Many KIs felt the response was nationally led when compared to the international-led Ebola response. An *"Ebola style"* institution was set up–the National COVID-19 Operations & Response Centre (NACOVERC), a multi-agency collaboration, where most members were from ministries (Health, Defence, Foreign Affairs etc)—that took *"operational"*, *"frontline"* and *"strategic"* leadership, supported by technical committees that provided scientific advice. Extensive news coverage of the international outbreak before the first case in Sierra Leone enabled early preparation and strong national-level response.

*Population health-measures.* Preventive measures put in place by the government included the cessation of international flights, restriction of intra-district travel, social distancing, face masks, handwashing on entry at public and business premises, reduced numbers seated on public transportation, lockdowns, and the use of a nightly curfew.

*"The government still emphasizes on putting on your facemask, social distancing, hand sanitiser and all those stuff people still do that."*

*(Senior Doctor)*

*New governance structures.* The national COVID-19 response, led by the NACOVERC, was a reactivation of the National Ebola Response Centre (NERC), an emergency structure designed for Ebola. The NERC had eight technical pillars [24] and the NACOVERC response utilised a similar adapted pillar structure: Surveillance, Case Management, Labs, Drugs and Medical Supply, Risk Communication and Social Mobilisation, Psychosocial, Food Assistance and Nutrition. NACOVERC aimed to coordinate via the pillars—national scale management, training, and harmonisation of health workers, policymakers, resources, financing, and public health messaging.

*"We work with NACOVERC, so our health workers are there, the Directorate of Health Security and Emergencies."*

*(KI)*

**9. Role of international organisations.** Financial support, technical advice and specialist teams were provided by international organisations to support the COVID-19 response, for example, to expand laboratory capabilities. However, this support was at a reduced scale in comparison to the Ebola outbreak.

*Supported us to take the lead.* Policy and planning were discussed by two KIs in detail. NACOVERC and the MoHS led the multi-agency national response compared to the international-led response during Ebola. The government valued the technical advice from international partners.

*'We have to commend the international partners. . . they have been wonderful their contributions are significant. Yes, maybe the difference is that they support us to take the lead, unlike Ebola where they took the lead.'*

*(KI)*

*Not as much financial support as Ebola.* Funds were generated and distributed across the health system through the nationally led response and with international support, enabling a swift response to COVID-19 when compared to Ebola.

*"By the time April ended we had started implementing that national plan, we started getting money from government and by the time June ended we were developing with the World Bank [separate funded] project."*

*(KI)*

Although NGOs provided some consumable donations e.g., rice and oil, international aid and funding were reported to be far lower than during the Ebola crisis by both health workers and policymakers.

*'I don't think we have enjoyed much of funding from international organizations. I have mention some specific support like from DFID [Department for International Development] which has been part of their ongoing supports and maybe CDC [Center for Disease Control] gives some support on some specific aspect like for data maybe surveillance, ICAP [formerly International Center for AIDS Care and Treatment Programs] and few, but nothing to compare with Ebola absolutely.'*

*(KI)*

### Patient experiences

**10. Patient pathway.** *No money to travel.* Barriers to accessing care included those related to high transportation costs to facilities and unavailability or reduced public transport services, especially during the lockdown. Private taxis were available but only *'people with money'* could use them.

*"Well, there was no transportation, especially district lockdown transportation was high, and the people with money travels only."*

*(Patient)*

Patients complained about an increase in the cost of drugs because of shortages. Customer services such as toilets were also closed in many facilities to comply with IPC measures negatively affecting their overall experience of accessing care.

*"Well one of the difference would have been in the cost of the drugs we bought. The drugs would not have cost us as much as it did [before COVID-19]"*

*(Patient)*

'Government hospital has no toilet facility they are there but the control measures is poor so they closed them'

*(Patient)*

*"We went there to do the [Covid] test but didn't do the operation there as a result of the cost. They said five million Leones. They do that type of operation, but the problem was the money"*

*(Patient)*

**11. Socio-economic impacts.** Due to public health measures (lockdowns, curfews, social distancing) some livelihoods and socialising were restricted with damaging impacts on communities due to rising costs of goods and transport to healthcare facilities. Savings were reported to be spent on food and shortages were further impacted by a lack of price control. This had an impact on patients' health-seeking behaviour with fewer funds available for transport, costs of treatment and drugs, and essentials like food during the hospital stay.

*Things have not been easy.* Patients reported their jobs were restricted or not permitted during different phases of the outbreak, causing financial strain on families and communities. Sales of non-essential items or food goods were reduced. Consequently, they had less money available for healthcare.

*"I was working but due to the corona I lost my job, all of us. Since the corona things have not been easy then the business too is not easy, the hardship, my brothers that were working are not working anymore."*

*(Patient)*

*Feeling isolated.* Public gatherings such as funeral rites and weddings and socialising in homes were restricted or highly limited meaning some people felt isolated.

*"We were not interacting, we were not visiting no one, even when you visit someone, he or she will tell you no go to your own house."*

*(Patient)*

*It's a lie.* While most respondents were aware of the pandemic and were adhering to national public health measures, some thought the whole pandemic was a lie and it did not exist.

*"People started denying that the sickness [COVID-19] doesn't exist, that it was just a lie. So, they were disobeying the law."*

*(Patient)*

**12. Learning from Ebola.** Interviews with health workers, patients, and policymakers reported perceived learning from Ebola that was implemented during COVID-19. People's

experience of Ebola reduced denial of COVID-19 and meant they were more supportive of measures to contain its spread.

*Not a new thing.* Health workers reported '*activating*' systems like IPC and triage, and the use of PPE, to manage COVID-19 which had been developed during Ebola. Many health workers were familiar with these measures and triage and IPC training were reinstated quickly. Even at the national level, similar task forces as were present during Ebola were reinstated to manage the COVID-19 national response. The language of the Ebola response: 'triage', 'screening', 'testing', 'isolation', 'infectious diseases unit', 'enhanced IPC', 'hand-washing', 'veronica bucket' and 'treatment centres', were familiar to people and created a shared understanding of the terms and actions required in relation to COVID-19.

> "*For corona when it came that experienced, we had from the Ebola it helped us, I mean how to put on PPEs is not a new thing that we need to learn, it just like a revision that refreshed our training*".
>
> *(Senior Surgeon)*

*Not as bad as Ebola.* Health workers reported being unprepared for Ebola but for COVID-19 they felt they were better prepared, mentally as well as in terms of taking practical measures at work. They attributed this to their previous experience and extensive reporting of the COVID-19 outbreak in other countries.

> "*Ebola took us unaware; we were not prepared for it. It just popped-up like that. The COVID-19 started somewhere, far away from Sierra Leone. We knew that it will come one day. So, we started preparing for it.*"
>
> *(Senior Doctor)*

The mechanism of transmission and treatment was reported amongst health workers to influence perceptions and comparison of mortality between Ebola and COVID-19. Health workers felt COVID-19 was less serious as the case fatality was lower than in Ebola and therefore people were more likely to present at the health facility. In the case of Ebola, people were more likely to hide their symptoms.

> "*The only thing like for Ebola because of the vomiting, the rapid way it manifest it seem very dangerous. Like for this corona if you come early, you will be treated it, but Ebola when you catch it you barely recover and most of the people that came with Ebola because they were hiding, they will bring a more hopeless situation and that made so many death occurs.*"
>
> *(Senior Matron)*

## Discussion

This study explored the perceptions and experiences of patients, health professionals and policymakers relating to the functioning of the Sierra Leonean health system during the country's first wave of COVID-19 in 2020. As anticipated, we found that all six core WHO health systems building blocks were impacted by COVID-19 as well as two additional themes which developed from people's narratives—'patient experience' and 'learning from Ebola'. Study

findings could inform future actions and policy by highlighting the strengths and areas for improvement related to the COVID-19 response which emerged during analysis.

There were four key findings within the study. The memory of and knowledge retained from the Ebola outbreak both for individuals and institutions had a significant, positive impact on their handling of COVID-19. The global nature of the pandemic and the resulting reduction in international support for the COVID-19 response in Sierra Leone led to increased national ownership, leadership, and action. The indirect effects of public health measures (rising living costs, lack of transport, lack of access to livelihoods) and people's perceptions of COVID-19 were barriers in accessing healthcare. Pre-existing issues within the healthcare system related to infrastructure and resources were compounded by the pandemic and these negatively affected the ability of health workers to address patient needs and the quality of care reportedly received by patients.

## How Ebola influenced the COVID-19 response

As evidenced by multiple examples—the positive effect of institutional and individual learning from living and working through the Ebola outbreak was significant and indicates sustained resilience within the health system. Institutional memory of the Ebola experience influenced national decision making and translated into policy and action; examples include the organisation of NACOVERC, set-up of Coronavirus Treatment Centres (CTCs) and Community Care Centres (CCCs), COVID-19 testing protocols and lockdown decisions. This is in stark contrast to the failures of leadership documented during Ebola [25] and reflects Wildavsky's (1988) [26] interpretation of resilience, defined as a form of learning that develops over time, in response to repeated crisis exposure, which strengthens future practice.

A multi-country study including West Africa, investigating the influence of organisational memory of previous epidemics on timely COVID-19 response, suggests that health system learning is associated with the implementation of earlier COVID-19 responses of between 6–10 days in all policy areas [27]. Many respondents cited the benefits of having time to plan and prepare for COVID-19 by watching global reporting in contrast to the lack of warning and speed of Ebola. One study focused on Sierra Leone—though limited to patients with HIV—found evidence of improved preparation for COVID-19 within health facilities because of learning from the mechanisms of Ebola (screening/triage, infection prevention and control, isolation) though this learning does not appear to have translated through to community services [28]. Public perception of COVID-19 was influenced heavily by familiar experiences during Ebola, public health measures were better recognised and understood even if they were not always adhered to.

Previous experience working during the Ebola outbreak helped health workers feel more confident and competent to organise and deliver COVID-19 care whilst safeguarding their own safety by enacting triage protocols and correctly using PPE. However, anxiety about resource gaps, being exposed, infected, and the death of colleagues because of COVID-19 during their work, was prevalent and was mirrored during Ebola [29]. Dean et al., 2020 [30] suggest that health workers exposed to repeated shocks may retain unresolved trauma caused by crises which is exacerbated by the fragility of low resource setting work. This is relevant for most health workers in Sierra Leone and highlights the importance of providing mental health support for them now, and in preparation for future health emergencies.

The reduced role of international organisations in Sierra Leones COVID-19 response was described as *"enabling"* the national response to lead itself; this sense of ownership is distinctly different to feelings of frustration and confusion during Ebola [29]. The trade-off in gaining national ownership was a severe reduction in international financial and technical support

because of the global nature of the pandemic with countries worldwide facing unprecedented health system challenges [31].

## Health system response

The use of volunteer workers [32] and non-payment or delay of salaries and incentives [33, 34] affecting morale is a recurring theme much written about both in times of crisis and normal functioning in Sierra Leone's health system [35, 36]. During COVID-19, multiple strikes [37, 38] related to remuneration were seen which negatively impacted services. Health workers expressed the psychological toll of risking their lives as well as the emotion of feeling devalued because of pay disputes. Although it was not discussed by interviewees, in 2020 the MoH did support some health workers in specific COVID-19 treatment settings by providing health insurance, life insurance, tax relief and extra payments *"hazard pay"*. These support packages sparked national debate and their implementation was delayed due to negotiations with professional associations representing health workers and verification exercises [38, 39]. Unfortunately, the provision of hazard pay for *"risky"* work with COVID-19 cases may have skewed HCW and facilities away from non-COVID care, having a detrimental effect on those patient groups.

Extra demands on staff time due to training or meetings related to COVID-19 caused rotas to be changed and meant higher workloads for those left covering clinical shifts. The creation of staffing shortages by external training in health emergencies should be considered and avoided. One way to achieve this may be employing training models that cascade learning and encourage staff to teach their teams in the workplace and shifting to on-job mentoring approaches; these were described in interviews as appreciated, valued, and offering elements of sustainability.

Both formal and informal communication channels remained important for HCW's to obtain information about COVID-19; there was a definite gap highlighted in national level updates being cascaded down to facility teams. Hospital planning and decision making centred on internal structures rather than updates from the MoHS or NACOVERC. These mechanisms could be strengthened to improve future emergency responses. Communication between health actors is considered a core attribute of health system resilience in emergencies [40] with an existing, proven leadership structure in place (well tested by Ebola and COVID-19) there is an opportunity to strengthen internal health system communication to support better coordinated and effective responses in future.

The response at community level was also supported by community health workers who served as social mobilisers and contact tracers by providing community education through proper communication in a culturally appropriate way (mirroring events during Ebola where most community health workers also played this role [41]). Additionally, during COVID-19, they worked in quarantine homes [42]. Again, as described during the Ebola response, the role of community engagement in communication, combating fear and building relationships of trust were crucial [43], as the efforts of community health workers directly impacted the numbers of patients accessing facilities.

## Patient experience—Accessing and receiving services

People did not report changing their health-seeking behaviour because they were fearful of being infected with COVID-19 within health facilities as was the case during Ebola [40] this contrasts with findings from Ghana [44] and the US [45] where fear of in-hospital COVID-19 infection was a key reason for not accessing care. However, the indirect effects of public health measures on trade and travel increased prices in many areas of life and affected people's ability

to earn a living. This financial barrier was the most frequently reported reason for not accessing care or accessing care and deciding not to proceed to treatment. In a World Bank survey [46] of 11 sub-Saharan African countries conducted early in the pandemic, 51% of respondents cited a lack of money as the dominant barrier to accessing healthcare.

Adherence to public health measures was mixed with some examples of families enforcing no-visit rules and others who did not comply—citing disbelief in COVID-19 altogether. During the pandemic COVID-19 denial and belief in virus origin conspiracy theories has been a problem in both high and low-income countries [47, 48]. It was suggested by some that the ban on public gatherings, nightly curfew and travel restrictions led to feelings of social isolation. This is well documented in the literature linking social distancing measures with a deterioration in mental health and resilience [49]. Some of the strategies suggested to combat isolation and loneliness during the pandemic such as contact with family wearing PPE, accessing formal/virtual mental health support services, using technology to enhance social connections, and making sure basic needs are met, are challenging for those with socioeconomic, internet connectivity and health service provision challenges [50, 51]. Other strategies such as taking part in outdoor activities and connecting with neighbours may be more feasible and culturally appropriate [52].

## Strengths and limitations

The strengths of this study are its reach and diversity. Geographically, narratives were captured from respondents living and working in both rural and urban environments and from health workers in large tertiary hospitals through to small peripheral health units. Interviews also represented a varied range of voices and perspectives from each level of the system from users (patients) to healthcare providers (encompassing a range of professions), to those involved in the COVID-19 response at the national level. Only interviewing patients who were able to come to health facilities, and so our findings do not capture the views, experiences and difficulties of people who did not or were unable to access facilities. This group was potentially even more vulnerable and faced increased challenges in accessing health care. This study also focused only on government facilities, the role of private and non-profit services within Sierra Leone's health landscape is important and was not captured. Fewer KIs were interviewed than planned due to challenges in securing interviews at the peak of the pandemic thus precluding theme saturation, although these KIs played an important role in the national response.

## Future research

We suggest issues that warrant further investigation are the perceptions and experiences of patients who did not reach health facilities during this time, the role played by private and non-profit providers within the healthcare ecosystem, and the longer-term impact of COVID-19 on the issues studied as the pandemic continues.

## Conclusion

In conclusion, when accessing healthcare during COVID-19, patients reported issues affecting the quality of the care provided such as service closures, delays, and the unavailability of medicine. The cost of transport and care, driven up by public health measures, was prohibitive and highlighted a clear divide between different socioeconomic groups and their ability to access healthcare.

Many of the human and system challenges reported in this study are not new, but they were amplified during the Ebola or COVID-19 crises. They reflect many underlying constant issues,

as summed up well in one quote: "*These are challenges that have not come in because COVID is here, these are chronic health system challenges that COVID has just exposed*".

## Supporting information

**S1 Checklist. Inclusivity in global health research questionnaire.** This questionnaire provides further information on the ethical, cultural, and scientific considerations specific to inclusivity in global research as they relate to this study.
(DOCX)

**S1 File. This document contains the interview guides used with patients, healthcare workers and KIs.**
(PDF)

## Acknowledgments

We would like to thank Fatmata Bah and Kadija Shaw, who conducted the interviews, and Esther Mansaray for providing administrative support. We are grateful to our respondents who provided their valuable time to speak to us. Finally, we express our gratitude to all front-line workers and communities who led the effort to combat COVID-19 in Sierra Leone.

## Author Contributions

**Conceptualization:** Haja Wurie, Andrew J. M. Leather, Justine I. Davies, Håkon A. Bolkan, Stephen Sevalie, Daniel Youkee, Divya Parmar.

**Data curation:** Haja Wurie.

**Formal analysis:** Hana Stone, Emma Bailey, Divya Parmar.

**Funding acquisition:** Andrew J. M. Leather.

**Methodology:** Haja Wurie, Andrew J. M. Leather, Justine I. Davies, Håkon A. Bolkan, Stephen Sevalie, Daniel Youkee, Divya Parmar.

**Project administration:** Divya Parmar.

**Writing – original draft:** Hana Stone, Emma Bailey.

**Writing – review & editing:** Hana Stone, Emma Bailey, Haja Wurie, Andrew J. M. Leather, Justine I. Davies, Håkon A. Bolkan, Stephen Sevalie, Daniel Youkee, Divya Parmar.

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
