## [Decision Letter · Decision Letter 0]

14 Aug 2023

PONE-D-22-20556“We plan because if you don’t plan, you plan to fail” - A qualitative study examining the health system’s response to COVID-19 in Sierra Leone.PLOS ONE

Dear Divya,

Thank you for submitting your manuscript to PLOS ONE. After careful consideration, we feel that it has merit but does not fully meet PLOS ONE’s publication criteria as it currently stands. Therefore, we invite you to submit a revised version of the manuscript that addresses the points raised during the review process.

We look forward to receiving your revised manuscript.

Kind regards,

Joseph Adu, PhD, MSc., Mphil

Academic Editor

PLOS ONE

Journal Requirements:

3. Please include a complete copy of PLOS’ questionnaire on inclusivity in global research in your revised manuscript. Our policy for research in this area aims to improve transparency in the reporting of research performed outside of researchers’ own country or community. The policy applies to researchers who have travelled to a different country to conduct research, research with Indigenous populations or their lands, and research on cultural artefacts. The questionnaire can also be requested at the journal’s discretion for any other submissions, even if these conditions are not met.  Please find more information on the policy and a link to download a blank copy of the questionnaire here: https://journals.plos.org/plosone/s/best-practices-in-research-reporting. Please upload a completed version of your questionnaire as Supporting Information when you resubmit your manuscript.

4. Please provide the full name of the ethics committee from Sierra Leone that approved your study.

Reviewers' comments:

Reviewer's Responses to Questions

**Comments to the Author**

1. Is the manuscript technically sound, and do the data support the conclusions?

Reviewer #1: Partly

Reviewer #2: Yes

2. Has the statistical analysis been performed appropriately and rigorously? 

Reviewer #1: N/A

Reviewer #2: N/A

3. Have the authors made all data underlying the findings in their manuscript fully available?

Reviewer #1: No

Reviewer #2: Yes

4. Is the manuscript presented in an intelligible fashion and written in standard English?

Reviewer #1: Yes

Reviewer #2: Yes

5. Review Comments to the Author

Reviewer #1: The topic of the study is relevant and interesting. However, the paper needs some revision in some of the sections.

1. It would be interesting to know the rationale for choosing surgical and maternity care as an indicator of overall health system functioning as both types of health services are for special populations.

2. The methodology section could be revised by including the following information:

a. How were the interview guidelines developed? Was it pilot tested on any participants?

b. How were the study participants approached? Was there a process to set up an appointment with the study participants for the interview?

c. How were the interviews with the study participants conducted? Was it in-person or via some virtual platforms like Zoom, WhatsApp, Teams?

d. How did the authors decide on the sample size of the study or how was the sample size deemed sufficient for the study?

e. Were the translated audios independently checked by the team members to ensure the reliability of the translation?

f. Please revise Table 2 as the numbers do not add up to a sample size of 24.

3. While the efforts of the authors to create Figure 1 is appreciable, Figure 1 can be decluttered to make it more visually effective.

a. Please add a note for any abbreviations used in the Figure.

Reviewer #2: The authors have presented an impactful and interesting study entitled: “We plan because if you don’t plan, you plan to fail” - A qualitative study examining the health system’s response to COVID-19 in Sierra Leone”. The authors may consider addressing the following comments to help improve the quality of the papers.

(1) The title may be revised as it may not be informative or clear to most readers.

(2) The abstract should clearly show the backgrounds (including aims), methods, results, and conclusion.

(3) Detailed information about the recruitment/interview should be provided. E.g. :

(a) How did authors ensure that participants understood the interview questions ?

(b) At what stage did authors achieved a thematic saturation ? .

(c) How the transcripts were coded ? ,

(d) Potential disagreement in the coding of the transcript, and how resolved if any ?

(4) In Line 151: Data Analysis, the information provided are very general. Authors needs to explain in detailed how was data going to be analysed? Need more detail here.

6. PLOS authors have the option to publish the peer review history of their article (what does this mean?). If published, this will include your full peer review and any attached files.

Reviewer #1: No

Reviewer #2: No

---

## [Author Response · Author response to Decision Letter 0]

28 Sep 2023

Point-by-point response uploaded as a separate file.

---

## [Decision Letter · Decision Letter 1]

2 Nov 2023

A qualitative study examining the health system’s response to COVID-19 in Sierra Leone.

PONE-D-22-20556R1

Dear Dr. Parmar,

We’re pleased to inform you that your manuscript has been judged scientifically suitable for publication and will be formally accepted for publication once it meets all outstanding technical requirements. Please, ensure that the manuscript is proofread for grammatical errors and typos to avoid undue delay while transferred to produuction. Also, endeavour to ensure that all references align with Plos One referencing stlye.

Kind regards,

Joseph Adu, PhD.

Academic Editor

PLOS ONE

---

## [Editor Report · Acceptance letter]

25 Jan 2024

PONE-D-22-20556R1 

PLOS ONE

Dear Dr. Parmar, 

I'm pleased to inform you that your manuscript has been deemed suitable for publication in PLOS ONE. Congratulations! Your manuscript is now being handed over to our production team.

Kind regards, 

on behalf of

Dr Joseph Adu 

Academic Editor

PLOS ONE